# *Ixodes scapularis* Is the Most Susceptible of the Three Canonical Human-Biting Tick Species of North America to Repellent and Acaricidal Effects of the Natural Sesquiterpene, (+)-Nootkatone

**DOI:** 10.3390/insects15010008

**Published:** 2023-12-22

**Authors:** Eric L. Siegel, Guang Xu, Andrew Y. Li, Patrick Pearson, Sebastián D’hers, Noel Elman, Thomas N. Mather, Stephen M. Rich

**Affiliations:** 1Laboratory of Medical Zoology, Department of Microbiology, University of Massachusetts, Amherst, MA 01003, USA; esiegel@umass.edu (E.L.S.); pbpearson@umass.edu (P.P.); 2United States Department of Agriculture, Agricultural Research Service, Invasive Insect Biocontrol and Behavior Laboratory, Beltsville, MD 20704, USA; andrew.li@usda.gov; 3Computational Mechanics Center, Instituto Tecnológico de Buenos Aires (ITBA), Ciudad Autónoma de Buenos Aires C1106ACD, Argentina; sdhers@itba.edu.ar; 4GearJump Technologies, Limited Liability Company, Brookline, MA 02446, USA; noel@gearjumptech.com; 5Center for Vector-Borne Disease, University of Rhode Island, Kingston, RI 02881, USA; tmather@uri.edu

**Keywords:** acaricide, natural repellent, nootkatone, ticks

## Abstract

**Simple Summary:**

Ticks are responsible for transmitting several disease-causing agents to humans and animals, including those causing Lyme disease and Rocky Mountain spotted fever. Human protection from tick bites relies on personal protection tactics for preventing or minimizing tick feeding. One common method is the use of chemicals for repelling or killing ticks. This study sought to assess the repellent capability of the tick-killing compound, nootkatone, which is found in natural sources like grapefruit oil, against adult ticks in a laboratory setting. Nootkatone can be found in two main structural arrangements in nature, resulting in the ability to distinguish between a (+) isomer and a (−) isomer. The (+) isomer has a lower odor threshold (and is therefore more potent in its fragrance) and was chosen for use in this study. Adult females of three tick species, the American dog tick (*Dermacentor variabilis*), the blacklegged tick (*Ixodes scapularis*), and the lone star tick (*Amblyomma americanum*), were repelled by nootkatone at different concentrations. Blacklegged ticks were most susceptible to this compound, not only being repelled at very low nootkatone concentrations, but also experiencing significant mortality when checked 24 h after exposure. American dog ticks and lone star ticks required much higher concentrations to repel and were not killed by the brief exposures. These experimental results indicate that nootkatone may be used in a dual action killing/repelling role against adult ticks, with particularly strong effects against blacklegged ticks.

**Abstract:**

Ticks are vectors of many human and animal zoonotic disease-causing agents causing significant global health and economic strain. Repellents and acaricides are integral to the human capacity for personal protection from tick bites. Nootkatone, a naturally occurring sesquiterpene found in the Alaskan cedar tree, grapefruit, and other sources, has been documented to be a potent acaricide. Research has also noted repellent effects against some tick species. In this study, our aim was to investigate the effect of synthetic, high-purity (+)-nootkatone on adult *Ixodes scapularis*, *Dermacentor variabilis*, and *Amblyomma americanum* ticks in an in vitro, vertical filter paper bioassay. (+)-nootkatone showed compelling tick repellency, but median effective concentrations (EC_50_) significantly differed among species. *Ixodes scapularis* were repelled at very low concentrations (EC_50_ = 0.87 ± 0.05 µg/cm^2^). Higher concentrations were required to repel *D. variabilis* (EC_50_ = 252 ± 12 µg/cm^2^) and *A. americanum* (EC_50_ = 2313 ± 179 µg/cm^2^). Significant post-exposure mortality, assessed 24 h after repellency trials, was also observed in *I. scapularis* but was absent entirely in *D. variabilis* and *A. americanum*. These tests demonstrate that nootkatone has a promising dual-action personal protection capacity against adult *I. scapularis* ticks, warranting further investigation in more natural environments and in the presence of host cues.

## 1. Introduction

Ticks pose a significant and growing burden to global health and economics [1,2]. Despite concerted efforts to develop effective control tactics, the geographical range of ticks and the incidence of tick-borne diseases continue to grow [3]. In the eastern United States, the blacklegged tick, *Ixodes scapularis* Say, transmits *Borrelia burgdorferi* sensu stricto, the Lyme disease spirochete that is primarily responsible for an estimated 476,000 yearly infections [4,5]. This tick also transmits other emerging pathogens for which humans are incidental hosts, including *Anaplasma phagocytophilum*, *Babesia microti*, *Borrelia miyamotoi*, and Powassan Type-2 virus [6,7,8]. The American dog tick, *Dermacentor variabilis* (Say), and the lone star tick, *Amblyomma americanum* (Linnaeus), are two other tick species that are abundant and expanding their range in the United States. *Dermacentor variabilis* is a vector for less prevalent disease agents, such as those causing Rocky Mountain spotted fever and tularemia [9]. *Amblyomma americanum* transmits several agents including *Ehrlichia chaffeensis*, Heartland virus, and Bourbon virus. Its bite is also associated with alpha-gal syndrome, a form of allergic sensitization to red meat due to an interaction with the carbohydrate epitope alpha-gal (galactose-alpha-1,3-galactose) in the tick salivary glands [10].

Personal protection measures are essential to reducing tick bite risks. These include methods that individuals can employ to reduce the chance of bites or limit the amount of time ticks feed, such as performing full-body tick checks, wearing acaricide-treated or untreated protective clothing, and taking showers shortly after outdoor activity [11,12,13]. One effective form of personal protection involves the treatment of clothing or skin with compounds that kill or repel ticks. The most prominently used compounds are permethrin, a type I synthetic pyrethroid clothing treatment that kills ticks, and DEET (*N*,*N-diethyl-m-toluamide*), a synthetic repellent that may be sparingly applied to the skin [14,15]. Though permethrin and DEET are the current gold standards for tick bite protection, both have limitations regarding user agreeability, application compliance, adverse health effects, and variable effectiveness against some tick species [16]. *Amblyomma americanum*, for example, is a remarkably fast-moving species and responds aggressively to host cues, such as carbon dioxide and heat. Consequently, these ticks have been noted to be less susceptible to barrier treatments on people and are a key target for innovative and effective alternatives [16].

Natural products based on essential oils for killing and repelling arthropods have garnered significant recent attention due to their potential as safe and environmentally friendly alternatives to synthetic formulations [17,18]. These natural products may exhibit less toxicity to humans and off-target plants and animals and be more pleasant in terms of fragrance and the feeling on the skin [17]. However, the repellency of these compounds is often questionable. A natural product that has potentially shown efficacy against a range of vectors is nootkatone, a naturally occurring eremophilane sesquiterpene found in plants, such as the Alaska yellow cedar tree (*Callitropsis nootkatensis*), the oil of grapefruit (*Citrus paradisi*), and vetiver grass (*Chrysopogon zizanioides*) [19]. It is widely used in the flavor and fragrance industries, carrying the characteristic aroma and taste of grapefruit [20]. Nootkatone is “generally regarded as safe” (GRAS) by the United States Food and Drug Administration (FDA) under the Federal Food, Drug, and Cosmetic Act, indicating that it may have the potential to serve in a personal protective role against arthropod vectors while being safe enough to eat [21]. It is also registered by the United States Environmental Protection Agency (EPA) as a biopesticide as of 2020 [22]. Nootkatone has been demonstrated to be repellent to a range of pathogen-transmitting insects, such as *Aedes aegypti* and *A. albopictus* mosquitoes, the principal vectors of chikungunya virus, Dengue fever, and Zika virus [23]. It also has been shown to have effects against the Formosan subterranean termite (*Coptotermes formosansus*), the common fruit fly (*Drosophila melanogaster*), and aphids (*Myzus persicae*; *Rhopalosiphum padi*) [24,25,26].

To date, most of the reported work has focused on nootkatone’s capacity as an acaricide environmentally applied to kill tick species, including *Rhipicephalus microplus* (Canestrini), *Rhipicephalus sanguineus* (Latreille), *Amblyomma cajennense* (Fabrcius), *Hyalomma lusitanicum* Koch, *Amblyomma americanum*, *D. variabilis*, and *I. scapularis* [27,28,29]. Nymphal ticks often receive the most attention in risk assessments since nymphal *I. scapularis* ticks are major vectors of Lyme disease spirochetes [1]. The adult stages of *A. americanum*, *D. variabilis*, and *I. scapularis* are commonly encountered on humans in the United States and present a substantial risk of diseases such as Powassan virus disease, anaplasmosis, ehrlichiosis, tularemia, Rocky Mountain spotted fever, and babesiosis [6]. While repellency conferred by nootkatone has been recognized in laboratory and field studies, nootkatone-based repellents are not currently marketed for personal protection against ticks [30,31,32]. Repellency against adult ticks is also critically understudied. The objective of this work was therefore to contribute to the fundamental understanding of (+)-nootkatone as a repellent against adult ticks by investigating the capacity of (+)-nootkatone, the isomer with favorable bioactivity and a favorable odor threshold, to repel *I. scapularis*, *A. americanum*, and *D. variabilis* adult, female ticks in an in vitro bioassay.

## 2. Materials and Methods

### 2.1. Tick Sourcing and Storage

*Ixodes scapularis* and *D. variabilis* adult, female ticks were collected via flag sampling in Western Massachusetts. This consisted of sweeping a light-colored cloth fixed to a flag handle through host-seeking tick-infested vegetation [33]. *Amblyomma americanum* ticks were collected in Eastern Massachusetts, USA. Ticks were collected in the spring of 2023. Field-collected ticks were removed from the flag with forceps and placed in 7-dram polystyrene vials with ventilated, cloth-fitted snap caps. A blade of grass was placed in the vial to maintain moisture. Once returned to the lab, ticks were stored at 4 °C for up to 24 h, with vials placed in an air-tight container containing a piece of wet towel to maintain moisture. Two hours before use, ticks were placed in an environmental chamber at 20 °C, 90% relative humidity (RH), and equilibrated at room temperature 5 min before testing. After trials, ticks were returned to holding vials in the environmental chamber for 24 h, before being checked for post-exposure mortality.

### 2.2. Preparation of Nootkatone Solutions

Technical-grade (+)-nootkatone (≥98%, Sigma-Aldrich, St. Louis, MO, USA) was used in the preparation of three stock solutions with ethanol (absolute, Thermo Fisher Scientific, Waltham, MA, USA) as a solvent: 40%, 10%, and 0.1%. These concentrations were used based on the results of informal preliminary testing with the same tick populations (not shown). Solutions were kept at 4 °C for up to 24 h before being discarded and making a fresh stock. These were diluted with ethanol (absolute) to obtain six working concentrations unique to each species (Table 1). Stock and test solutions were stored in 13 mm × 100 mm tubes with screw caps (Simport Scientific, Saint-Mathieu-de-Beloeil, QC, Canada); they were vortexed with a VWR fixed-speed vortex mixer (VWR, Radnor, PA, USA) for 30 s when initially made and again before use.

### 2.3. Repellency Bioassay

A vertical filter paper bioassay was modified from previously developed tick repellency tests [16,34]. Whatman No. 4 filter paper (test card) was cut into a 4 cm × 7 cm rectangle. Lines were drawn across the test card 1 cm from either end, creating a 4 cm × 5 cm box in the center. The working solution (165 µL) was applied to the center region of the test card. The test card was allowed to dry for 10 min before being suspended on a glass stir rod with a small alligator clip. The temperature and humidity were 20 °C and 50% RH throughout the trials, which were performed on a laboratory bench.

The 1500 adult, female ticks used in the repellency trials (600 *I. scapularis*, 600 *D. variabilis*, and 300 *A. americanum*) were split equally into the 6 concentrations and were tested individually (one tick per test card); 100 *I. scapularis*, 100 *D. variabilis*, and 50 *A. americanum* were used per concentration. An additional 20 ticks of each species were used in solvent-only control groups. A brief test for inclusion/exclusion was performed to ensure that selected ticks climbed the card. Ticks were placed at the bottom of an untreated card. Ticks that climbed up or showed random movement were included. Those that did not climb or dropped from the card were excluded. Ticks were taken directly from this test and placed at the bottom, untreated space of the test card (either treated or solvent-only control) using clean forceps. Trials lasted a maximum of 5 min. Ticks that reached the top or dropped were removed, and the trial was immediately concluded. Ticks were repelled if they remained in the bottom, untreated region of the card or dropped from the card without reaching the top, untreated surface. Ticks were not repelled if they reached the top, untreated surface of the card. Repellency in this context was therefore defined by the absolute prevention of ticks crossing a 5 cm long treated surface within a 5 min period. After being stored in an environmental chamber after exposure for 24 h at 20 °C and 90% RH, they were assessed for mortality by observing body posture, appendage movement, and response to thermal and respiratory stimuli emitted by the observer.

### 2.4. Data Analysis

Data were analyzed in Minitab, version 21.4 (Minitab, Inc., State College, PA, USA) [35]. A probit model was used to make sigmoid dose–response curves and obtain concentrations (adapted from experimental surface densities) that repel 50% (median, EC_95_) and 95% (EC_95_) of subjects with fiducial confidence intervals (Appendix A) [36]. Model fit was assessed with a deviance chi-squared test; *p* > 0.05 indicated an acceptable fit. A confidence interval overlap test assessed significant differences in effective concentrations [37]. Non-overlapping confidence intervals of EC_50_ were considered significant differences (*p* < 0.05). The significance of post-exposure mortality was assessed relative to controls with a chi-squared test for differences in proportions [36].

## 3. Results

### 3.1. Controls

The solvent-only treatment did not impact the behavior of any control ticks. All 20 ticks of each species successfully reached the top of the card by the end of their respective trials. *Ixodes scapularis* and *A. americanum* showed the greatest tendency to climb, whereas some *D. variabilis* often exhibited random movement around the filter paper before either settling at the top or escaping the top by climbing onto the clip holding the card after some time. All were noted to be alive 24 h after exposure.

### 3.2. Nootkatone Repellency Trials

Nootkatone demonstrated clear dose-dependent repellency with significant differences in efficacy by species (Figure 1, Table 2). Repelled ticks dropped from the card while climbing on the treated surface or withdrew when approaching the treated surface. Probit models showed acceptable fit for each species, though fit was marginally better for *A. americanum* (*x*^2^ = 2.525) and *I. scapularis* (*x*^2^ = 3.719) than it was for *D. variabilis* (*x*^2^ = 7.236) (Table 2). With an estimated EC_50_ at 0.87 µg/cm^2^ [95% CI: 0.82–0.92], nootkatone was most effective against *I. scapularis.* At the highest concentration (4.13 µg/cm^2^), 100% repellency was achieved. Significantly higher concentrations were required to repel *D. variabilis*, EC_50_ = 252 µg/cm^2^ [95% CI: 240–264], and *A. americanum*, EC_50_ = 2313 µg/cm^2^ [95% CI: 2134–2492] (EC_50_ overlap *p* < 0.05). All *D. variabilis* were repelled at a concentration of 825 µg/cm^2^. Only 72% of *A. americanum* were repelled at 3300 µg/cm^2^, the highest concentration of nootkatone tested.

### 3.3. Tick Mortality

Significant reductions in the survival of *I. scapularis* were observed at the four highest concentrations ranging from 0.52 to 4.13 µg/cm^2^, as assessed 24 h after trials (Figure 2). Exposure to the third-highest concentration, 1.03 µg/cm^2^, resulted in the most significant post-exposure mortality (63%), and this dropped slightly with higher concentrations, with 4.13 µg/cm^2^ leading to only 25% mortality. In contrast, all *D. variabilis* and *A. americanum* ticks were alive when assessed (Figure 2).

## 4. Discussion

This study evaluated the in vitro repellency of (+)-nootkatone against adult female ticks of three species (*I. scapularis*, *D. variabilis*, and *A. americanum*) using a vertical filter paper bioassay. The repellency of nootkatone has been understudied for adult-stage ticks in previous research. Human–adult tick encounters are common and can result in the transmission of disease-causing agents [1,6,38,39]. The importance of adult ticks in agent transmission dynamics differs by species. In the case of *I. scapularis*, it is very clear that nymphs are the primary vectors of the causative agents of Lyme disease, anaplasmosis, and babesiosis, based on the seasonal occurrence of disease cases relative to the seasonal host-seeking activity of nymphs and adults [1]. However, small peaks of these diseases occur in the fall, consistent with adult activity in the northeastern and upper midwestern United States [6]. For *D. variabilis*, the nymphal stage is rarely encountered in nature, and more than 99% of ticks removed from humans are adults. *Amblyomma americanum* adults and nymphs have an overlapping questing season and are removed from humans in relatively similar numbers. These examples reinforce the notion that tests against adults are important in the product development process [39].

(+)-Nootkatone was repellent to *I. scapularis* adult females at very low concentrations but significantly less effective against *D. variabilis* and *A. americanum*. Even at high concentrations (40%, 3300 µg/cm^2^), only 70% of *A. americanum* ticks were repelled. This suggests that a commercial repellent product may need a high nootkatone concentration as a base for adult ticks. However, it is not uncommon to see products with high concentrations of active ingredients. DEET, for example, is often found to be formulated at concentrations of 40–98%. Nevertheless, cost must be considered with solutions containing higher concentrations of new active ingredients. It would take 400,000 grapefruits to make 1 kg of nootkatone, so it is instead produced through synthetic means at a higher price point than that of other active ingredients like DEET [40].

Active ingredients, formulations, and delivery systems that can target multiple tick species and life stages are becoming particularly relevant with expanding tick ranges, resulting in the overlapping geographic presence of several tick species [3,41,42]. Species-specific responses to repellents are well documented in ticks. For example, *I. scapularis* has been noted to be generally more susceptible to repellents, though studies often yield different results, and direct comparison between studies is difficult [43]. Multiple variables may impact tick response even when the same species or life stage and repellent formulation are used. These include bioassay design, genetic differences within tick populations, tick age and storage conditions, and the working definition of repellency, which is largely unstandardized [44].

Particularly important variables for nootkatone-specific comparisons pertain to active ingredient sourcing, formulation, and purity [45]. Nootkatone can be extracted from a variety of natural (grapefruit and cedar trees) or synthetic sources (synthesized from nootkatone precursors) at a variety of purities, with differing relative isomer compositions. The volatile fragrance threshold of the (+) isomer, used in this study for example, is approximately 1000 times lower than that of the (−) isomer, which may be found in naturally occurring nootkatone at varying concentrations [46]. Solutions containing greater amounts of (−)-nootkatone would therefore require much higher concentrations to achieve comparable effects of a (+)-nootkatone equivalent. Many formulations (crude, emulsified, and microencapsulated) have also been described that result in differing exposure conditions and persistence on the treated surface [47,48]. The variation in repellency observed from these differences makes interstudy comparisons of nootkatone repellency particularly difficult and indicates that qualitative comparisons may be of the most value.

A study of various fractioned nootkatone compounds against *I. scapularis* nymphs determined that effective repellent nootkatone concentrations range from 3.3 to 6.6 µg/cm^2^ [32]. Another study noted reductions in tick retention on infested coveralls using a 10% spray-on solution with slightly higher efficacy against *I. scapularis* than *A. americanum*, which appears to be consistent with the results of the present study [31]. Only one study has considered nootkatone repellency against *D. variabilis* adult ticks, which demonstrated an increase in the time to cross a treated barrier with 1090 µg/cm^2^ of synthetic nootkatone, notably close to the higher bound of efficacy observed in this study [30]. Future studies are also needed to better understand the differences in nootkatone repellency across species and life stages; however, collective results suggest that nootkatone may function as an efficacious repellent against immature and adult ticks of a number of species.

DEET, the gold standard to which novel products are compared, was not assessed in this study and therefore direct comparisons of efficacy with nootkatone cannot be made. We may, however, speculate on performance based on previous studies to guide future work. *I. scapularis* nymphs have been shown to be more repelled by DEET (EC_95_ = 4.25 µg/cm^2^) than *A. americanum* (41.6 µg/cm^2^) [16]. Adult *I. scapularis* have been shown to be less repelled by DEET than nymphs, with a test showing that 12.9 µg/cm^2^ is able to repel 62% of a combination of males and females (Li, unpublished). This concentration is greater than that used in the present study for *I. scapularis*, suggesting that (+)-nootkatone may exhibit higher efficacy than DEET against *I. scapularis* adults. Male ticks were not used in this study as females are known to bite hosts and feed to repletion, while males are more relevant for reproduction and may bite less often [49]. Future studies could assess sex-stratified responses to nootkatone as males are still important in the reproductive maintenance of tick populations and thus important to the maintenance of stable transmission cycles [50].

The present study confirms that nootkatone has a dual action capacity (repelling and killing) against *I. scapularis* adult ticks. Although the vertical filter paper bioassay was not originally designed to assess mortality, significant numbers of *I. scapularis* died within 24 h after exposure, indicating that only very brief contact at low concentrations was actually lethal. Moderate concentrations (1.03 µg/cm^2^) proved the most lethal. Exposure to the highest concentration, 4.13 µg/cm^2^, in comparison, resulted in a higher frequency of ticks falling off during their trials. It was also evident that not all ticks physically came into contact with the treated surface, as many walked up to the treated line and would not proceed, suggesting that non-contact repellency may play a role in these effects. This was most evident with *I. scapularis* and less so with *A. americanum*.

Nootkatone extracts of Alaska yellow cedar in concentrations as low as 0.0029%, lower than the repellent concentrations in this study, have been shown to kill *I. scapularis* adult ticks under laboratory conditions [51]. *Amblyomma americanum* have been shown to be less susceptible to nootkatone’s lethal effects, consistent with the reduced repellent efficacy and lack of mortality observed in this study [27]. The vertical filter paper bioassay design must be noted to have likely inflated the effective concentrations for repellency and to have reduced the chance of observing mortality with *A. americanum* and *D. variabilis.* These metastriate species, in comparison to the prostriate *I. scapularis*, are larger and move very quickly. Given exposure in a different setting—such as in larger arenas or with direct application (sprayed directly on to the tick)—a higher efficacy of repellency and lethality would likely be observed, based on what has been shown in previous studies [16]. It is unknown how long these species must contact a repellent-treated surface before responding. Beyond this, there are several unknowns in behavioral ecology regarding the responses to odorants and tick questing behavior across species, such as the willingness to ignore the presence of a repellent for “ambushing” ticks compared to more actively “hunting” ticks. Careful consideration of the dual action of nootkatone must be made in practice since effective killing may be prevented by effective repellency at suboptimal concentrations.

The language used to describe repellency when applied to ticks is often ambiguous due in part to the lack of standardized means to describe and measure how these active ingredients modify tick behavior [52]. Active ingredients with effects including lethality, contact irritation, spatial repellency, and others are sometimes collectively referred to as repellents, though these do not always reflect the traditional definition of repellency (oriented movement away from a treated source). Personal protection compounds used against mosquitos and other insects that spend mere seconds on a host typically act on odorant receptors, disrupting host cue detection through spatial repellency effects or preventing landing through an irritant effect [53]. The host seeking and feeding behaviors of ticks are very different from that of insects and requires different means of measure [44]. Ticks quest by climbing vegetation or other vertical surfaces to wait for a passing host on which to grab and navigate up the host in search of a feeding site, where they remain attached for several days [54]. The EPA considers this behavior in their recommendations for in vitro testing via the human arm bioassay, where repellency is defined by ticks crossing 3 cm upwards into a treated region after being released to walk from an untreated surface [44]. However, we can speculate that distinct events occur during the initial interaction with a protected human, which may affect the tick response. These events have not been defined, but in the simplest configuration, would pertain to whether the initial encounter of the host by the tick is with a repellent-treated surface (skin or clothing) or an untreated surface followed by an encounter with a repellent-treated surface while navigating towards a bite site. Obviously, the most effective repellents would prevent tick bites under both scenarios. The dynamics of the initial tick–host interaction are poorly understood, and the distinction between protective efficacy and these means requires further study.

The present in vitro bioassay takes advantage of the innate tendency of ticks to climb vertical surfaces. We define repellency as the interruption of that climbing behavior. This is repellency in the strict sense and may be directly analogous to the observed repellency caused by contact with DEET or picaridin, wherein the tick makes an oriented movement away from or falls from treated surfaces. In the case of DEET and picaridin, repellency is presumed to be due to an irritant/deterrent and spatial repellency combination [16]. Further work is needed to assess the range of tick behaviors that can be targeted by nootkatone, such as expellency (detachment) in the case of existing infestations at the time of treatment, or impact on the success of feeding measured via molting and egg mass laying success [51].

Nootkatone’s lethal effect against insects has been attributed to a picrotoxin-like mechanism of GABA_A_ receptor channel antagonism; however, no work has been conducted to identify a tick-killing mechanism [24]. Repellency mechanisms in insects and ticks also are yet to be characterized [50,55]. This knowledge gap is largely due to our minimal understanding of tick chemosensory physiology, including the mechanisms behind the reception of most active ingredients. Against insects, DEET is known to inhibit olfactory neuron receptors and acetylcholinesterase, thus producing a spatial repellent response that masks host odors [52]. Recent studies have identified olfactory recognition pathways in the tick that result in spatial repellency [56,57]. Thus, multiple mechanisms are responsible for the repellency we observe. Similar effects have been shown with SS220, indalone, and the pyrethroid spatial repellents transfluthrin and metofluthrin [58,59]. The assay used in the present study cannot differentiate between gustatory, touch, and olfactory effects due to the limited scope of physical observation; however, distinguishing between reluctance to contact a treated surface and readily walking on a treated surface before impact is important [58]. Future work may employ other designs to further explore these effects in more detail [56,59]. Differences in nootkatone’s effects between species should also be further investigated. *Amblyomma americanum* are larger, faster, and more aggressive than *I. scapularis*, and are very sensitive to the olfactory cues (heat and carbon dioxide) emitted by hosts. This difference in activity is particularly interesting because if *A. americanum* relied more on olfactory cues relative to *I. scapularis*, one may expect that *A. americanum* would be more readily repelled. Instead, we often see the opposite trend. There are vast unknowns in our collective knowledge regarding the chemosensory physiology of these species that require much more investigation to reveal.

In its raw form, nootkatone is highly volatile. Its short residual activity (due to vapor loss, surface absorbance, and inactivation via photolysis) reduces effective contact with ticks and subsequent repellent/acaricidal effects [60]. Future research is needed to explore how nootkatone can be best formulated and delivered for consumer and environmental use. Further, though it is safe for human use and with off-target hosts including bees and mammals, nootkatone is phytotoxic and may have adverse effects on plants at higher concentrations [46]. Multiple avenues of work have begun addressing these concerns [47,48]. Lignin-based encapsulation has been demonstrated to provide stronger and longer-lasting acaricidal activity, reductions in phytotoxicity, and longer environmental persistence compared to emulsified and crude formulations [48]. The controlled release device integration of compatible formulations has yet to be explored but also may have promising answers for these applications [61,62,63,64,65]. The parent structure of nootkatone has also provided the opportunity for manipulation with its many points of structural diversity, resulting in the synthesis of derivatives with demonstratively improved insect antifeedant activity and reduced phytotoxicity and volatility [26]. A formulation integrating advanced delivery techniques with more chemically active derivatives may perform better than the volatile and crude ethanol-based formulation used in this study.

## 5. Conclusions

Nootkatone was demonstrated to repel *I. scapularis*, *D. variabilis*, and *A. americanum* in an in vitro vertical filter paper bioassay with concentrations that significantly differed between species. This work builds on previous studies identifying repellency with nootkatone against some tick species. Future studies are needed to assess recently developed formulations and novel nootkatone derivative performance such as those of dual-action (killing and repelling) environmental treatment and personal protective products against ticks.

## Figures and Tables

**Figure 1 insects-15-00008-f001:**
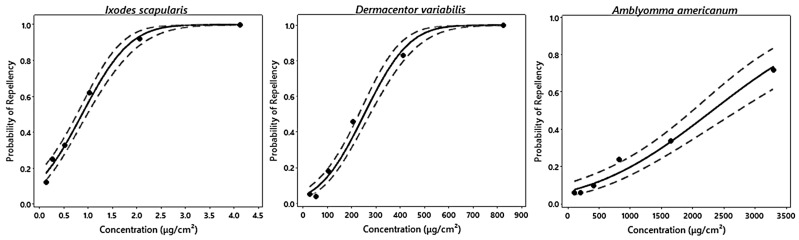
Binary fitted line plots and associated 95% confidence intervals of repellency observed in nootkatone trials across 6 experimental concentrations separated by species: *Ixodes scapularis*; *Dermacentor variabilis*; and *Amblyomma americanum*.

**Figure 2 insects-15-00008-f002:**
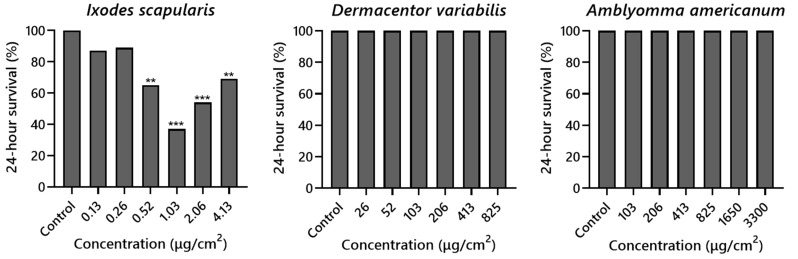
Figure showing 24-h post-exposure survival of *Ixodes scapularis*; *Dermacentor variabilis*; and *Amblyomma americanum*. Significant reductions in survival for active ingredient-inclusive trials relative to the solvent-only control are noted, based on a chi-squared test for differences in proportions: ** *p* < 0.01; *** *p* < 0.001.

**Table 1 insects-15-00008-t001:** Solutions of nootkatone used in repellency trials and corresponding surface density once applied to test cards.

Species	Solution (*w*/*v*, %)	Surface Density (µg/cm^2^)
*Ixodes scapularis*	0.05	4.13
	0.025	2.06
	0.0125	1.03
	0.00625	0.52
	0.003125	0.26
	0.0015625	0.13
*Dermacentor variabilis*	10	825.00
	5	412.50
	2.5	206.25
	1.25	103.13
	0.625	51.65
	0.3125	25.78
*Amblyomma americanum*	40	3300.00
	20	1650.00
	10	825.00
	5	412.50
	2.5	206.25
	1.25	103.13

**Table 2 insects-15-00008-t002:** Effective concentrations of (+)-nootkatone repellency against adult females of each tick species.

Species	*n*	EC_50_ (µg/cm^2^) ± SE ^1^	EC_95_ (µg/cm^2^) ± SE	Probit Fit *x*^2^ (*p* Value) ^2^
*Ixodes scapularis*	600	0.87 ± 0.05	2.15 ± 0.13	3.719 (0.445)
*Dermacentor variabilis*	600	252 ± 12	491 ± 26	7.236 (0.124)
*Amblyomma americanum*	300	2313 ± 179	4864 ± 439	2.525 (0.640)

^1^ EC_50_ values significantly differed (*p* < 0.05) between species as assessed with an overlap test of confidence intervals. ^2^ Probit model goodness-of-fit based on a deviance chi-squared test with four degrees of freedom. *p* > 0.05 indicated acceptable fit.

## Data Availability

All data presented in this study are found within the manuscript.

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
