# Peer review of "Ixodes scapularis Is the Most Susceptible of the Three Canonical Human-Biting Tick Species of North America to Repellent and Acaricidal Effects of the Natural Sesquiterpene, (+)-Nootkatone"

_insects, 2023, doi:10.3390/insects15010008_

Round 1

Reviewer 1 Report

Comments and Suggestions for Authors

This paper investigates the repellency effect of (+)-nootkatone against three adult tick species in laboratory conditions, offering valuable insights into the potential use of this compound as a tick repellent. The paper is well-written and structured. However, in the Material and Method section (2.3), additional details are needed to enhance the reproducibility of the study. For instance:

Please specify the surface on which the ticks were placed when applying the solutions (e.g., petri dish, lab bench).

Clarify the size of the middle zone to which the solutions were applied.

Explain whether ticks were tested individually on separate cards or if groups of ticks were placed together on a single card.

Provide information on how the control group was treated.

Mention the location where the tests were performed, such as an incubator or another specific environment. This will provide a complete understanding of the experimental conditions.

In Table 2, it would be beneficial to add a column for the probit model to enhance the clarity of the presented data.

Author Response

This paper investigates the repellency effect of (+)-nootkatone against three adult tick species in laboratory conditions, offering valuable insights into the potential use of this compound as a tick repellent. The paper is well-written and structured. However, in the Material and Method section (2.3), additional details are needed to enhance the reproducibility of the study. 

The authors thank the reviewer for the feedback. Revisions are highlighted in the manuscript where noted. We feel the feedback has strengthened the manuscript. 

Please specify the surface on which the ticks were placed when applying the solutions (e.g., petri dish, lab bench).Mention the location where the tests were performed, such as an incubator or another specific environment. This will provide a complete understanding of the experimental conditions.

  1. Please refer to the graphical abstract and section 2.3, line 149 for a description of the assay including the surface of application. In addition, it has been clarified that the experiment was performed on a laboratory bench, line 155. 

Clarify the size of the middle zone to which the solutions were applied.

  1. Please refer to section 2.3, line 151 for a description of the size of the middle zone. 

Explain whether ticks were tested individually on separate cards or if groups of ticks were placed together on a single card.

  1. Please refer to line 158. Additional language was added to clarify. 

Provide information on how the control group was treated.

  1. Please refer to line 160 for a description of the control handling. 

In Table 2, it would be beneficial to add a column for the probit model to enhance the clarity of the presented data.

  1. Please refer to the last column for information on probit model fit in addition to footnote 2. “Model fit” has been changed to “probit fit” to improve clarity. 

Reviewer 2 Report

Comments and Suggestions for Authors

Re: mdpi manuscript Insects-2757172

INTRO: Note that the “generally regarded as safe” (GRAS) designation for materials is generated by the U.S. Food & Drug Administration (FDA), not the U.S. EPA. The U.S. EPA has registered nootkatone as a “Biopesticide”, which is also an important designation.

Throughout: The term “in vitro” applies when research is conducted with grinding and binding studies, e.g., literally “in glass”. Use the term “in vivo” when conducting experiments that use whole living organisms; it literally means “in living”. These experiments are done with whole living organisms. 

Line 173: The probit model used here produces sigmoid dose-response curves; the more conventional way of determining the responses and EC50 values is to use a log-probit model, which produces straight lines as responses. The manuscript’s models are not wrong, just less conventional; they do not need to be changed in this manuscript.

Lines 197-199: The calculated EC50 values and C.I.’s should not be presented to 5 significant figures. For example, 251.50 µg/cm2 cannot be defended as accurate, given the margins of error in all of the measuring, etc. and other techniques used in this study; the authors cannot be confident of the accuracy of any figures beyond 3 figures. Use 0.87, 251, and 2,313 µg/cm2 for the EC50 values. Furthermore, the extra figures are distracting (besides inaccurate), and the 0.87, 251, and 2,313 are much easier for the reader to assimilate and remember.

Author Response

The authors thank the reviewer for the constructive comments and attention to detail.

INTRO: Note that the “generally regarded as safe” (GRAS) designation for materials is generated by the U.S. Food & Drug Administration (FDA), not the U.S. EPA. The U.S. EPA has registered nootkatone as a “Biopesticide”, which is also an important designation.

The mention of EPA as the authority designating GRAS status has been fixed to FDA, see line 94 highlights. 

Throughout: The term “in vitro” applies when research is conducted with grinding and binding studies, e.g., literally “in glass”. Use the term “in vivo” when conducting experiments that use whole living organisms; it literally means “in living”. These experiments are done with whole living organisms. 

The authors thank the reviewer for raising this query. For tick repellency studies, in vitro studies involve a controlled setting to observe their reactions to the chemical without involving a host. In vivo studies involve the presence of a host, human or otherwise in the assay.

The assay used in this study would be considered an in vitro repellency bioassay.

Please refer to the following recent publication which evaluated some of these assays: Burtis JC, Ford SL, Parise CM, Foster E, Eisen RJ, Eisen L. Comparison of in vitro and in vivo repellency bioassay methods for Ixodes scapularis nymphs. Parasit Vectors. 2023 Jul 10;16(1):228. doi: 10.1186/s13071-023-05845-7. 

Line 173: The probit model used here produces sigmoid dose-response curves; the more conventional way of determining the responses and EC50 values is to use a log-probit model, which produces straight lines as responses. The manuscript’s models are not wrong, just less conventional; they do not need to be changed in this manuscript.

A supplemental figure has been included to show the linear response from the probit model (not shown in the original manuscript) based on the transformation from the sigmoid dose-response curve. Probit and logit models barely differed from each other but probit model was chosen as it was the best fit. 

Lines 197-199: The calculated EC50 values and C.I.’s should not be presented to 5 significant figures. For example, 251.50 µg/cm2 cannot be defended as accurate, given the margins of error in all of the measuring, etc. and other techniques used in this study; the authors cannot be confident of the accuracy of any figures beyond 3 figures. Use 0.87, 251, and 2,313 µg/cm2 for the EC50 values. Furthermore, the extra figures are distracting (besides inaccurate), and the 0.87, 251, and 2,313 are much easier for the reader to assimilate and remember.

The authors thank the reviewer for this point on clarity. The description of effective concentrations has been amended accordingly in each location necessary. 

Reviewer 3 Report

Comments and Suggestions for Authors

The manuscript focused on an in vitro experiment that assessed nootkatone repellency against Ixodes scapularis, Dermacentor variabilis, and Amblyomma Americanum adults. It also assessed mortality. The manuscript was well written and informative. It provides more evidence on the efficacy of a promising compound for biorational tick management. It needs only minor revisions and is recommended for publication. Suggestions for improvement are below:

Title: The title could be improved as it doesn’t quite capture all aspects of the manuscript. Repellency is the focus here, but mortality was also a substantial (and important) component. Maybe focus on the main point that I scapularis repellency and susceptibility was superior to the other two species.

Line 16: ‘disease’ to ‘disease-causing’

Line 20: ‘of a’ to ‘of the’

Line 61: In numerous places, sentences begin with an abbreviated genus name. (also line 75, 119, 121, 185, 322 and maybe elsewhere). The genus name should be spelled out when at the start of sentences.

Line 120: Add “USA” after Massachusetts and what year were they collected?

Line 216 – Nootkatone could be deleted after experiment because its already mentioned in line 215.

Author Response

The manuscript focused on an in vitro experiment that assessed nootkatone repellency against Ixodes scapularis, Dermacentor variabilis, and Amblyomma Americanum adults. It also assessed mortality. The manuscript was well written and informative. It provides more evidence on the efficacy of a promising compound for biorational tick management. It needs only minor revisions and is recommended for publication. Suggestions for improvement are below:

The authors thank the reviewer for the report. The feedback has strengthened the manuscript. Revisions are highlighted in green. Requests for changes have been addressed in their entirety. 

Title: The title could be improved as it doesn’t quite capture all aspects of the manuscript. Repellency is the focus here, but mortality was also a substantial (and important) component. Maybe focus on the main point that I scapularis repellency and susceptibility was superior to the other two species.

  1. The authors agree that a revised title is more descriptive. We have changed it accordingly. 

Line 16: ‘disease’ to ‘disease-causing’

  1. We have revised this in all mentions.

Line 20: ‘of a’ to ‘of the’

  1. We have revised this text. 

Line 61: In numerous places, sentences begin with an abbreviated genus name. (also line 75, 119, 121, 185, 322 and maybe elsewhere). The genus name should be spelled out when at the start of sentences.

  1. Genus name at the beginning of sentence is now spelled out in all occurrences. Please refer to instances of green highlighted text. 

Line 120: Add “USA” after Massachusetts and what year were they collected?

  1. This has been revised.

Line 216 – Nootkatone could be deleted after experiment because its already mentioned in line 215.

  1. This has been revised, thank you.